# Electrospinning Ag-TiO_2_ Nanorod-Loaded Air Treatment Filters and Their Applications in Air Purification

**DOI:** 10.3390/molecules25153369

**Published:** 2020-07-24

**Authors:** Shan-Jiang Wang, Xiao-Yang Zhang, Dan Su, Yun-Fan Wang, Chun-Meng Qian, Xin-Ru Zhou, Yi-Zhi Li, Tong Zhang

**Affiliations:** 1Joint International Research Laboratory of Information Display and Visualization, School of Electronic Science and Engineering, Southeast University, Nanjing 210096, China; 230159658@seu.edu.cn (S.-J.W.); zxycom@seu.edu.cn (X.-Y.Z.); 220191474@seu.edu.cn (Y.-F.W.); 213170083@seu.edu.cn (C.-M.Q.); 213170614@seu.edu.cn (X.-R.Z.); 213170091@seu.edu.cn (Y.-Z.L.); 2Suzhou Key Laboratory of Metal Nano-Optoelectronic Technology, Suzhou Research Institute of Southeast University, Suzhou 215123, China; jssysls@163.com

**Keywords:** air pollution, air treatment filters, electrospinning, plasmon, hot carriers

## Abstract

The efficient treatment of the problem of air pollution is a practical issue related to human health. The development of multi-functional air treatment filters, which can remove various kinds of pollutants, including particulate matter (PM) and organic gases, is a tireless pursuit aiming to address the actual needs of humans. Advanced materials and nano-manufacturing technology have brought about the opportunity to change conventional air filters for practical demands, with the aim of achieving the high-efficiency utilization of photons, a strong catalytic ability, and the synergetic degradation of multi-pollutants. In this work, visible-responding photocatalytic air treatment filters were prepared and combined with a fast and cost-effective electrospinning process. Firstly, we synthesized Ag-loaded TiO_2_ nanorod composites with a controlled size and number of loaded Ag nanoparticles. Then, multi-functional air treatment filters were designed by loading catalysts on electrospinning nanofibers combined with a programmable brush. We found that such Ag-TiO_2_ nanorod composite-loaded nanofibers displayed prominent PM filtration (~90%) and the degradation of organic pollutants (above 90%). The superior performance of purification could be demonstrated in two aspects. One was the improvement of the adsorption of pollutants derived from the increase of the specific surface area after the loading of catalysts, and the other was the plasmonic hot carriers, which induced a broadening of the optical absorption in the visible light range, meaning that many more photons were utilized effectively. The designed air treatment filters with synergistic effects for eliminating both PM and organic pollutants have promising potential for the future design and application of novel air treatment devices.

## 1. Introduction

Air pollution triggered by various pollutants, such as particulate matter (PM) [1,2] and volatile organic compounds (VOCs) [3,4,5], is receiving extensive attention from researchers across the world [6]. Up to now, tremendous efforts have been made to solve this burning problem. Photocatalysis is deemed to be a green, environmental, and energy-saving technology in dealing with these environmental pollutants [7,8,9]. In practical use, wide bandgap photocatalysts, including TiO_2_, ZnO, etc., are usually regarded as a suitable medium featuring reliable photocatalyst activity and a lower toxicity and are cost-effective [10,11]. However, due to the limit of the width of the bandgap, these photocatalysts can only be activated by photons in the ultraviolet range, which unfortunately poses unpredictable damage to human health [12]. Meanwhile, the fast recombination of the photogenerated electron-hole pairs also restricts the photocatalytic capability [13,14]. Studies of novel photocatalysts with both improvement of the utilization of optical energy in the visible range and separation of electron-hole pairs are invariably meaningful in the air treatment field. In virtue of the plasmon-induced hot-electron effect [15,16,17], breakthroughs in the limits of catalytic ability can be applied to conventional photocatalysts. On the other hand, it has been found that the ability to capture photons in the visible range can be improved based on the surface plasmon resonance effect [18,19]. Additionally, the formed interfacial barriers can efficiently promote the separation of electron-hole pairs [20,21]. In particular, in recent work, researchers found that the efficiency of the utilization of hot-electrons was expected to reach the level of being practical, offering a prominent potential in the field of photocatalysis [22]. In addition, aiming at the practical demand, studies for effectively supporting photocatalysts and easier integration with common air purifiers have also been crucial [23,24]. Electrospinning-induced large-scale polymer nanofibers are gradually coming into the eyesight of researchers for their features of being porous, lightweight, flexible, and highly adsorbable [25,26,27]. Besides, nanofibers made by low-cost polymers can also be fabricated on a large scale by choosing suitable parameters of devices [28]. Moreover, they are regarded as having practical potential for replacing conventional routes of air filters [29,30,31]. However, there are still mechanisms and routes that need to be explored for rational strategies to improve the photocatalytic performance, including photocatalytic materials, filters, and the design of the synergetic mechanism, which will further promote development in future practical applications.

In this article, we propose large-scale electrospinning air treatment filters loaded with visible-responding photocatalysts, which can efficiently adsorb particulate pollutants and degrade VOCs simultaneously. Firstly, we synthesized Ag-TiO_2_ nanorod composites with a high-yield by simple photo-reduction routes, and the optical absorption was extended to the visible range with the doping of plasmonic nanostructures. Then, in order to realize the effective adsorption of PM particulates, improve the surface area for purification, and provide reliable supports, the three-dimensional, porous, and large areas of polystyrene nanofibers were spun, and the Ag-TiO_2_ nanorod composites were uniformly loaded by the process of automatic brushing. Finally, we made a simulation of real conditions for air purification, which showed superior air particulate filtering and toluene degradation. This work may contribute to the future development of novel air cleaners that are environmentally friendly and have a high efficiency.

## 2. Result and Discussion

### 2.1. Characterization of Ag-TiO_2_ Nanorod Composites and Electrospinning Nanofibers

Visible-responding Ag-TiO_2_ nanorod composites were synthesized by a multi-step reduction method mediated by the ultrasonic wave and UV light. As shown in Figure 1, TiO_2_ nanorods with a controlled size distribution were synthesized by a solvothermal method [32]. Then, the transparent mixture with AgNO_3_ in oleylamine and TiO_2_ nanorods in toluene was placed under an ultrasound bath for a short period of time so that Ag seeds were generated on the surface of nanorods. After that, the mixture was irritated by a UV light for different time periods up to 300 min. The products were collected by repeated centrifugation with absolute ethanol and then dispersed in toluene.

Figure 2a,b shows TEM images of the acquired TiO_2_ nanorods with a rod-shape and narrow size distribution (3 × 32 nm). The slow hydrolysis of titanium salt under an inert ambient and controllable heating-up process promoted them to form an anisotropic rod-shape instead of nanospheres. The yield of TiO_2_ nanorods could reach up to near 90% with a uniform morphology on a gram-scale. Figure 2c–h shows typical results of Ag-TiO_2_ nanorod composites after the continuous illumination of UV light (Sample 1–3 for 60, 180, and 300 min of illumination, respectively). When the time of illumination was 60 min, it can be seen that there was a single Ag nanoparticle attached to one TiO_2_ nanorod, and no Ag nanoparticle formed individually in the solution. The size of Ag nanoparticles was ~5 nm. When the time of illumination was raised to 180 min, much more Ag^+^ in the solution was utilized so that the number of Ag nanoparticles was increased on the surface of TiO_2_ nanorods. High-resolution TEM showed that there were several small Ag particles loaded on each TiO_2_ nanorod. The size of Ag particles ranged from 5 to 8 nm. However, the situation was a little different after a prolonged illumination period (above 180 min). The size of Ag nanoparticles was remarkably increased and individual particles were present in the mixture, as shown in Figure 2g,h. The tendency of forming larger Ag nanoparticles may be ascribed to a competitive growth mechanism [33,34], in which unstable and active small particles were dissolved to Ag^+^ by photogenerated oxidative species and then reduced to Ag atoms to be redeposited on larger and much more stable Ag nanoparticles.

The extinction spectrum also showed differences in diverse growing periods. As shown in Figure 3, pure TiO_2_ nanorods displayed weak optical absorption in the range of 400–900 nm. Additionally, the main optical absorption was located in the UV range corresponding to the bandgap of TiO_2_ nanorods. The extinction peak near 450 nm originated from the plasmonic effects of Ag nanoparticles. For samples 1–3, there was no remarkable shift of the peak during the 0–300 min of illumination, but the intensity of the peak was increased. Obviously, such changes could be rooted in the inevitable scattering effects of the larger Ag nanoparticles formed in the solution. Moreover, it could be seen that there was broad absorption of visible light, and the edge of the band could even reach 900 nm (see the inset of Figure 3). The extension of the optical absorption range could be attributed to the synergetic size effects and coupling effects of plasmonic Ag nanoparticles.

To achieve highly efficient air pollution degradation, we propose a nanofilter-loaded photocatalytic treatment configuration. Firstly, the air treatment filter was fabricated by an electrospinning process (Figure 4a). We chose polystyrene (PS) as an element as it can be easily acquired and is low-cost. Additionally, PS displays both chemical and thermal stability compared to other macromolecular materials. Figure 4b shows a microscopic image of the copper grid-loaded PS nanofibers. The PS nanofibers were firmly fixed on the substrate due to a strong electrostatic force. The average diameter for PS nanofibers was 150 ± 50 nm (Figure 4b). The multi-layer, large pores of the nanofilters enabled them to achieve a superior performance for removing PM particulates produced by the burning of incense (Figure 4c). After this, we successfully loaded the Ag-TiO_2_ nanorod composites on the PS nanofilters by automatic coating with a brush, as shown in Figure 4d. Based on this approach, the Ag-TiO_2_ nanorod composites were rapidly homogeneously fixed on the nanofilters, which were aroused by the synergistic effect of solvent evaporation and the Laplace pressure [35].

### 2.2. Photodegradation and PM Particulate Filteration Test

The test of photodegradation and PM particulate filtration was conducted in a shedding condition, which was used to simulate a real context. As shown in Figure 5a,b, a multi-functional air treatment filter was placed on the bottom of LED arrays, and a fan was situated nearby for promoting adsorption and degradation. PM particulates were generated by a commercial incense and calculated by a PM-VOCs monitor. Toluene was used as the main pollutant for photocatalytic degradation with the excitation of LED arrays.

Figure 6 shows the integrated capacity of filtration of PM particulates and the degradation of organic pollutants based on such filters. In Figure 6a, it can be seen that most PM particulates were filtered after a short period in pure fibers, TiO_2_-loaded fibers, and Ag-TiO_2_ nanorod composite-loaded filters. Ag-TiO_2_ nanorod composite-loaded filters displayed the best result (up to 90%) compared to the other two (~80% for TiO_2_-loaded fibers and ~60% for pure fibers). This can be ascribed to the improvement of the surface area and roughness after the loading of Ag-TiO_2_ nanorod composites, which also increases the contact areas, improving the catalytic performance. To verify the effectiveness of Ag-TiO_2_ nanorod composites, the effects of the degradation of toluene in three kinds of filters were also demonstrated. As shown in Figure 6b, the results demonstrated that both types of catalysts displayed the ability to decompose toluene in a limited duration. The Ag-TiO_2_-loaded fibers displayed a superior ability to degrade toluene in 120 min (above 90%), in contrast to TiO_2_-loaded fibers (20%).

The better performance exhibited by Ag-TiO_2_ nanorod composites could be summarized in two aspects (Figure 7). First, there was a remarkable improvement of the optical absorption in a wide range compared to TiO_2_ nanorods. The loading of Ag particles improved the ability to trap photons in a longer wavelength range induced by the plasmon resonance of Ag particles, which broke through the limit of the bandgap of conventional semiconductor-based catalysts. Secondly, the interfacial Schottky barriers hold back the backward transfer of generating electrons, which promotes the separating and utilization efficiency of hot carriers. Furthermore, a large surface area and high capacity of adsorption resulted from both electrospinning filters and photocatalytic materials further strengthened the performance of the air treatment filters.

## 3. Conclusions

In this work, Ag-TiO_2_ nanorod composite-loaded PS nanofiber air treatment filters have been shown to conduct the synergetic filtration of PM particulates and photodegradation of organic gases efficiently. The superior effects of visible catalysis were mediated by plasmon-induced composites. The photons could even be utilized in a range to 900 nm with the doping of plasmonic Ag, avoiding the limits of bandgaps of TiO_2_ and improving the usage of optical energy. The 3D, transparent, and large-area air filters were fabricated by fast electrospinning combined with low-cost PS. The strong intrinsic capacity of adsorption of nanofibers promoted the interaction between photocatalysts and fibers and the filtration of PM particulates. The decomposition of toluene could reach 90%, and the efficiency of the removal of PM could exceed 90%, at least in a very short time. For future industrial applications, multi-functional air treatment filters may have the potential to provide an optional route for multi-functional filters.

## 4. Materials and Methods

### 4.1. Synthesis of Silver/Titanium Dioxide Nanorod Composites

Silver/titanium dioxide (Ag-TiO_2_) nanorod composites were synthesized by previous reports, with minor modifications [36]. Firstly, TiO_2_ nanorods were synthesized by a solvothermal method, in a typical process, where oleic acid (0.14 M) was heated in a reflux condensation system under nitrogen protected at 160 °C for 1 h and was then cooled down to 30 °C undisturbedly. After that, a 0.02 M titanium source of tetrabutyl titanate (TBOT) was quickly injected, and the obtained mixture was heated to 280 °C and maintained for 3 h to form a solid rod-shape. After that, the solution was cooled down to 80 °C naturally. Excess absolute ethanol was added to stop the reaction, and the color was turned to brownish-yellow. The product was collected by repeated centrifugations with absolute ethanol and then re-dispersed in toluene.

Ag-TiO_2_ nanorod composites were synthesized under a fast and cost-effective route of UV-mediated photo-reduction. The oleylamine-dispersed silver nitrate (0.1 M), TiO_2_ nanorods in toluene (20 mg/mL), and 10 mL of toluene were mixed and stirred thoroughly in open-air for 1 h, and the transparent mixture with a faint yellow color was then placed under an ultrasonic source for 5 min to form Ag nanoclusters. Next, the solution was irradiated by a UV lamp for different periods. After that, a brownish black solution was obtained, and the precipitates were collected by centrifugation with absolute ethanol several times. The final product was re-dispersed in toluene for filter fabrication.

### 4.2. Synthesis of Polystyrene Nanofibers Loaded with Silver/Titanium Dioxide Nanorod Composites

The large-area, uniform air treatment filters were fabricated by electrospinning. In a typical process, 10 *w*/*w* % polymer solution was obtained by dispersing polystyrene (PS) (MW = 2.0 × 10^7^ g mol^−1^, Sigma-Aldrich) in dimethylformamide (DMF). Then, the transparent polymer solution was loaded in a 5-mL syringe with a 23-gauge needle tip. After that, the syringe was placed, and the collecting device was wrapped with copper grids with a wire diameter of 0.011 inches and mesh size of 18. The applied potential was 20 kV, the pump rate was 1 mL/h, and the time of electrospinning was 1 h. The final product was dried at 80 °C overnight in an oven.

The modification of Ag-TiO_2_ nanorod composites was conducted by a brush-painting method [35]. Briefly, the liquid Ag-TiO_2_ nanorod composites were smeared on the surface of filters in a 3D controlled direction by programmable software. After the solvent had completely evaporated, accelerated by a hot plate, the Ag-TiO_2_ nanorod composites were firmly fixed on the surface of filters.

### 4.3. Characterization

The morphologies and microstructures were characterized by transmission electron microscopy (TEM, Fei Tecnai T20) and field emission scanning electron microscopy (FE-SEM, Zeiss Ultra Plus, 15 kV). Extinction spectra were measured using a fiber optic spectrometer (PG 2000, Ideaoptics Technology Ltd. China). The ultrasonic process was conducted by an ultrasonic source (KQ5200DE, 200 W). The light sources were an ultraviolet lamp (UV, 365 nm, 24 W) and light-emitting diode (LED, 460 nm). The concentrations of PM particulates and toluene were monitored by a commercial tester (Siemens).

### 4.4. Filtration and Photodegradation Measurement

In this work, the source of PM particulates was derived from the burning of incense, and the size of PM particulates ranged from hundreds of nanometers to several micrometers. A PM monitor was used to demonstrate the performance of filters. The efficiency of filters (*η*) for eliminating PM particulates was calculated as follows [37]:(1)$$\eta =\left(1-\frac{C_{down}}{C_{up}}\right)\times 100\%,$$
where *C_down_* is the concentration of PM particulates after passing through the filters and *C_up_* is the concentration of PM particulates before passing through the filters.

Toluene (0.05 L) was chosen to evaluate the capacity of photodegradation of Ag-TiO_2_ nanorod composite-loaded filters. The concentration of toluene was calculated by a commercial tester. The efficiency of degradation was defined as follows:(2)$$T=\left(1-\frac{C_{after}}{C_{before}}\right)\times 100\%,$$
where *C_after_* is the concentration of toluene gases after catalytic reactions on the filters and *C_before_* is the concentration of toluene gases before catalytic reactions on the filters.

In a typical process, the prepared filter (50 cm^2^) was placed in a box. A small fan and an LED source (central wavelength of 460 nm) were also placed in the box. The fan was used for promoting the airflow of PM particulates and toluene, and the light source was used for driving the photocatalytic reactions. All of the reactions were mediated under an airtight and light-proof condition. The PM and VOCs monitor was pasted on the wall of the box to calculate the concentration of pollutants before and after the purification. All of the experiments and measurements were conducted under room temperature (27 ± 0.5 °C).

## Figures and Tables

**Figure 1 molecules-25-03369-f001:**
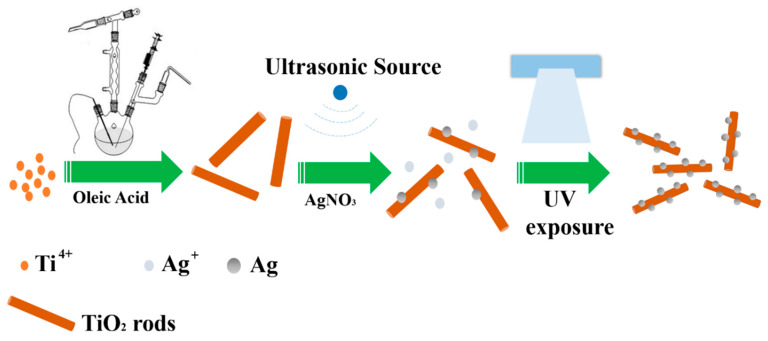
Scheme of the fabrication of Ag-TiO_2_ nanorod composites.

**Figure 2 molecules-25-03369-f002:**
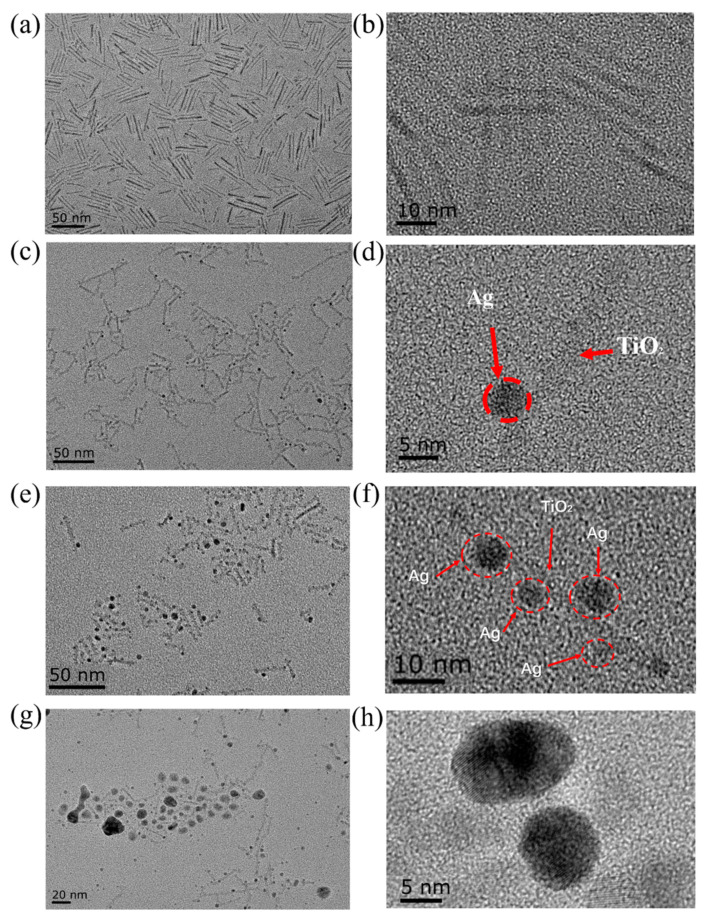
Characteristics of TiO_2_ nanorods and Ag-TiO_2_ nanorod composites. (**a**) TiO_2_ nanorods and (**b**) a high-solution image. (**c**) Ag-TiO_2_ nanorod composites with 60 min illumination of UV light (Sample 1) and (**d**) a high-solution image. (**e**) Ag-TiO_2_ nanorod composites with 180 min illumination of UV light (Sample 2) and (**f**) a high-solution image. (**g**) Ag-TiO_2_ nanorod composites with 300 min illumination of UV light (Sample 3) and (**h**) a high-solution image.

**Figure 3 molecules-25-03369-f003:**
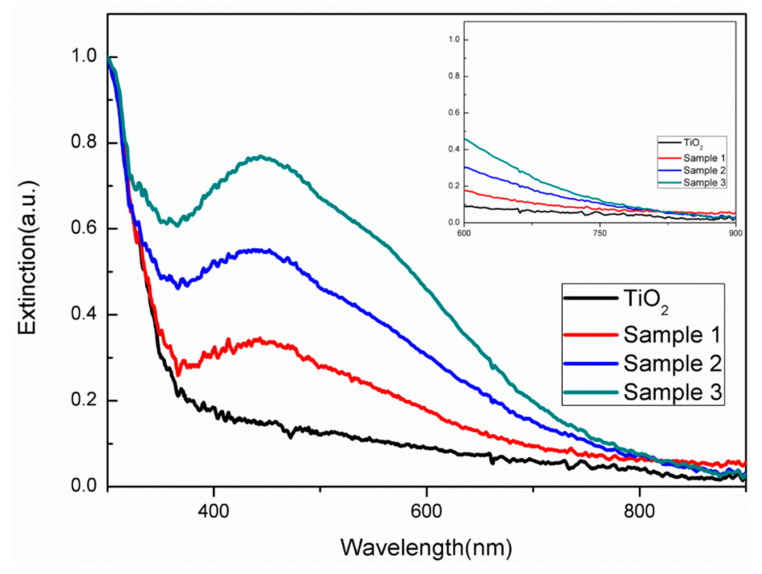
Extinction spectrum of pure TiO_2_ nanorods and Ag-TiO_2_ nanorod composites under the different times of illumination (Sample 1–3 with 60, 180, and 300 min of UV radiation). Inset shows the edges of TiO_2_ and Sample 1 to 3.

**Figure 4 molecules-25-03369-f004:**
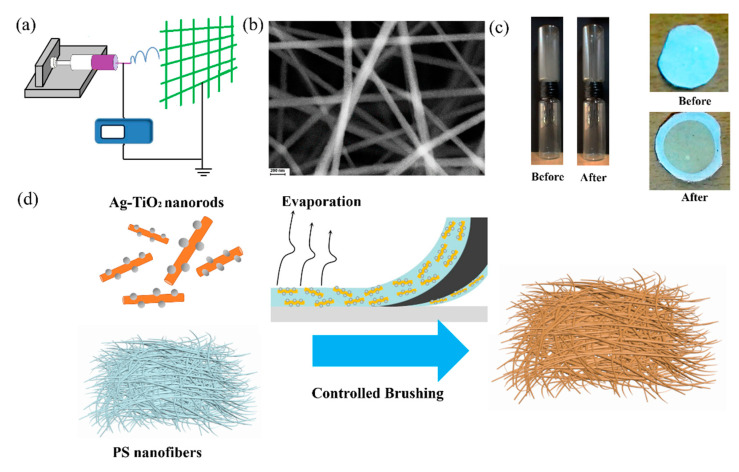
(**a**) A scheme of the process of electrospinning and (**b**) an SEM image of polystyrene (PS) nanofibers. (**c**) An experiment on the adsorption of PM conducted by using PS nanofibers. (**d**) The process of the fabrication of Ag-TiO_2_ nanorod composite-loaded PS nanofibers.

**Figure 5 molecules-25-03369-f005:**
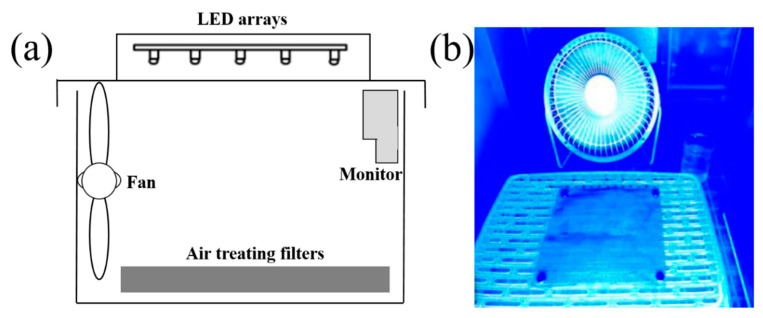
(**a**) The simulative air treatment devices and (**b**) a photograph of them under a working condition.

**Figure 6 molecules-25-03369-f006:**
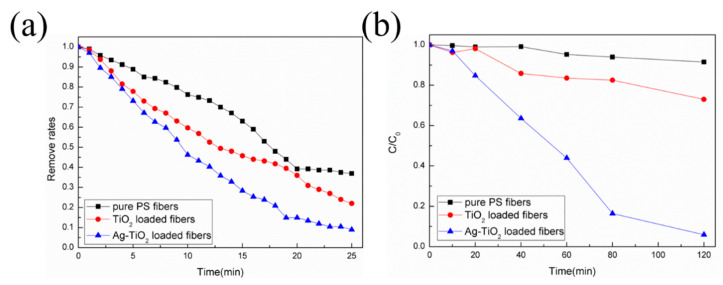
Characteristics of the performance of air treatment nanofibers. (**a**) The removal rates of particulate matter (PM) for three kinds of filters and (**b**) the efficiency of the degradation of toluene in such filters.

**Figure 7 molecules-25-03369-f007:**
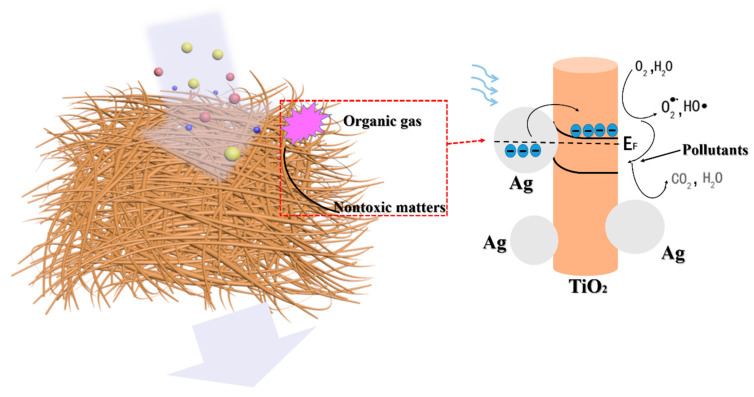
Mechanism for the degradation of air pollutants for the Ag-TiO_2_ nanorod composite-loaded PS nanofiber filters.

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
