# Peer review of "Electrospinning Ag-TiO_2_ Nanorod-Loaded Air Treatment Filters and Their Applications in Air Purification"

_molecules, 2020, doi:10.3390/molecules25153369_

Round 1
Reviewer 1 Report
This paper describes the combination of Ag nanoparticles on TiO2 nanorods that are supported on a porous polyethylene mesh network. This composite structure was examined for particle adsorption and toluene degradation via photocatalysis. With minor clarifications and revisions, this work should be acceptable for publication.
Minor grammar issues: line 13: pollution. line 47: "be applied to" instead of "be broke in"? line 49: "plasmon". line 50: "on the other hand" is used in two sequential sentences. line 51: "near recent" could be just "recent". line 85: should "under inertia" be "under inert"?
line 20: is there any data in the paper to suggest that this is a "cost-effective" process?
line 23 (and 131): the term "chinese brush" is not a term that I am familiar with so it may not mean much to the international readers.
line 93: It is unclear what the "more dissociative Ag+" statement means.
line 120: The inset figure's graphical data do not seem to match the larger graph's data. The large graph has blue and green lines crossing red at just above 800 nm, but the inset has blue crossing green below 700 nm and never cross under the red line.
line 164: I don't think doping of Ag particles occurs here (aren't the Ag nanoparticles on the TiO2 surface?) and the term "broaded" is unclear.
line 199: how much added toluene is "a certain amount"
Reviewer 2 Report
In my opinion, the proposed manuscript is devoted to the technical solution of a specific problem. I agree that this problem is of really high importance from the ecological point of view; and the proposed way can meet an interest from the industry.
Concerning the scientific meaning, I see nothing which can be consider as novelty, or surprised development of the problem in question.
On the other hand, there is shown one of the possible and effective way to solve the formulated problem; and at the same time, the manuscript contains no wrong statements or incorrect results (only, the using of the term "synergetic effect" seems a little questionable).
Thus, if the editorial policy allows one to publish such technical researches being only of application importance, the proposed manuscript can be recommended for publication in the Applied Chemistry Section.
